

# The Dark Cube: dark character profiles and OCEAN

Danilo Garcia[1,2,3,4] and Fernando R. González Moraga[2,5]

[1] Blekinge Center of Competence, Blekinge County Council, Karlskrona, Sweden
[2] Network for Empowerment and Well-Being, Sweden
[3] Department of Psychology, University of Gothenburg, Gothenburg, Sweden
[4] Department of Psychology, Lund University, Lund, Sweden
[5] Barn- och ungdomshabiliteringen, Region Kronoberg, Växjö, Sweden

## ABSTRACT

**Background**. The Big Five traits (i.e., openness, conscientiousness, extraversion, agreeableness, and neuroticism: OCEAN) have been suggested to provide a meaningful taxonomy for studying the Dark Triad: Machiavellianism, narcissism, and psychopathy. Nevertheless, current research consists of mixed and inconsistent associations between the Dark Triad and OCEAN. Here we used the Dark Cube (*Garcia & Rosenberg, 2016*), a model of malevolent character theoretically based on Cloninger's biopsychosocial model of personality and in the assumption of a ternary structure of malevolent character. We use the dark cube profiles to investigate differences in OCEAN between individuals who differ in one dark character trait while holding the other two constant (i.e., conditional relationships).

**Method**. Participants ($N = 330$) responded to the Short Dark Triad Inventory and the Big Five Inventory and were grouped according to the eight possible combinations using their dark trait scores (M, high Machiavellianism; m, low Machiavellianism; N, high narcissism; n, low narcissism; P, high psychopathy; p, low psychopathy): MNP "maleficent", MNp "manipulative narcissistic", MnP "anti-social", Mnp "Machiavellian", mNP "psychopathic narcissistic", mNp "narcissistic", mnP "psychopathic", and mnp "benevolent".

**Results**. High narcissism-high extraversion and high psychopathy-low agreeableness were consistently associated across comparisons. The rest of the comparisons showed a complex interaction. For example, high Machiavellianism-high neuroticism only when both narcissism and psychopathy were low (Mnp vs. mnp), high narcissism-high conscientiousness only when both Machiavellianism and psychopathy were also high (MNP vs. MnP), and high psychopathy-high neuroticism only when Machiavellianism was low and narcissism was high (mNP vs. mNp).

**Conclusions**. We suggest that the Dark Cube is a useful tool in the investigation of a consistent Dark Triad Theory. This approach suggests that the only clear relationships were narcissism-extraversion and psychopathy-agreeableness and that the malevolent character traits were associated to specific OCEAN traits only under certain conditions. Hence, explaining the mixed and inconsistent linear associations in the Dark Triad literature.

Corresponding author
Danilo Garcia,
danilo.garcia@icloud.com

## INTRODUCTION

Dark Triad Theory indicates that people's malevolent character consists of three traits: Machiavellianism, subclinical narcissism, and subclinical psychopathy (*Paulhus & Williams, 2002*). Machiavellianism is characterized by cynicism, manipulativeness (*Jones & Paulhus, 2009*), a cynical worldview, and lack of morality (*Christie & Geis, 1970*), narcissism is characterized by a tremendous sense of grandiosity, exploitativeness, and exhibitionism but, at the same time, a vulnerable self-esteem (*Morf & Rhodewalt, 2001*), thus having problems with criticism (*Raskin & Hall, 1979*), and psychopathy is characterized by low empathy, low conscientiousness, low anxiety, and high impulsive and high thrill-seeking behavior (*Furnham, Richards & Paulhus, 2013*; *Hare, 1985*). Although the Dark Triad traits have one thing in common, namely unagreeableness (*Garcia et al., 2015*; *Garcia & Rosenberg, 2016*; *Kajonius et al., 2016*), these malevolent character traits are suggested as overlapping but distinctive enough to warrant separate measurement (*Paulhus & Williams, 2002*). Subclinical studies have, for example, used personality models, such as the Big Five, to give a comprehensive view of these malevolent traits (*González, 2015*). The Big Five is a group of fundamental dimensions of personality often shortened as OCEAN: openness to experience, conscientiousness, extraversion, agreeableness, and neuroticism (*Costa Jr, McCrae & Dye, 1991*). These five relatively independent dimensions of personality are suggested to provide a meaningful taxonomy for studying individual differences (*John & Srivastava, 1999*; see also *Lee & Ashton, 2013*).

Individuals who score high in any of the three Dark Triad traits score low in agreeableness, individuals who score high in psychopathy and narcissism score high on extraversion and openness, while individuals high in Machiavellianism and psychopathy score low in conscientiousness (e.g., *Paulhus & Williams, 2002*). These associations are in line with a unified view of the dark traits, that is, suggesting at least a common unagreeable core for the three traits (*Jakobwitz & Egan, 2006*; *Paulhus & Williams, 2002*; *Garcia & Rosenberg, 2016*). Nevertheless, while some researchers have confirmed these results using different samples (e.g., *Lee & Ashton, 2005*), other researchers have not (e.g., *Jakobwitz & Egan, 2006*). At the multivariate level, the Big Five traits together seem to account for between 18% and 39% of the variance in the Dark Triad traits, again indicating only a moderate amount of overlap between OCEAN and the Dark Triad (see *Vernon et al., 2008*). In other words, even if there are some correlations between the Dark Triad and the Big Five, these are neither large nor consistent, except for agreeableness (*Vernon et al., 2008*).

These inconsistencies complicate the further exploration of the Dark Triad as a theory because the Dark Triad has not shown reliable correlations with available models (*Veselka, Schermer & Vernon, 2011*). Additionally, some researchers indicate that the three dark traits load on a single factor that explains 64% of variance in the traits (*Lyons & Rice, 2014*; see also *Garcia & Rosenberg, 2016*; *Kajonius et al., 2016*, who suggested a dyad instead of a ternary structure). That being said, most research has used linear assumptions between the two models at hand (i.e., the Dark Triad and the Big Five). Personality is instead better understood as a dynamic complex adaptive system (see among others *Cloninger, 2004*). Essentially, not all individuals who score high in psychopathy might score high in, for

example, extraversion and high levels of extraversion might lead to different expressions of malevolent character, that is, both psychopathy and/or narcissism. Moreover, the extrovert behavior of an individual high in both psychopathy and extroversion might differ depending on her/his level of Machiavellianism and narcissism. In other words, seeing personality as a dynamic complex adaptive system entails a person-centered approach in which an individual is not only adapting to the environment through her/his malevolent behavior, but also to the traits within the being–that is, the notion of the individual as whole system unit which is best studied by analyzing patterns of information or profiles (*Bergman & Wångby, 2014*). Although at a theoretical level there is a myriad of probable patterns of combinations of individuals' levels of dark character traits, if viewed at a global level, there should be a small number of more frequently observed patterns or "common types" (*Bergman & Wångby, 2014*; *Bergman & Magnusson, 1997*). Indeed, the development of character is best explained by nonlinear dynamics in complex adaptive systems that have led to a triune model or a character cube (*Cloninger, Svrakic & Svrakic, 1997*).

In this train of thought, *Garcia & Rosenberg (2016)* have presented an analogy to Cloninger's character cube (*Cloninger, 2004*), the dark cube, as a model of malevolent character to investigate conditional correlations by comparing, for example, OCEAN traits between individuals who differ in one dark trait while holding the other two constant (e.g., a profile characterized by *high* levels of Machiavellianism/low levels of narcissism/high levels of psychopathy vs. a profile characterized by *low* levels of Machiavellianism/low levels of narcissism/high levels of psychopathy). The character cube proposed by Cloninger has its basis on a biopsychosocial theory of human personality, which suggests the development of human personality as a result of the development of different regions in what has become the human brain. This research suggests that human character has a ternary structure: self-directedness (self-concept), cooperativeness (concept of relations with others), and self-transcendence (concept of our participation in the world as a whole) (*Cloninger & Garcia, 2015*; *Garcia et al., 2017b*; *Garcia et al., 2017c*; *Garcia et al., 2017a*; *Cloninger, 2007*; *Garcia et al., 2017d*). More than 30 years of research have confirmed the nonlinear dynamics of personality development, such as equifinality and multifinality,[1] and that the stepwise development of character determines large differences between individuals in their risk of psychopathology, as well as varying degrees of maturity and health that are best conceptualized as eight character profiles or the character cube (*Cloninger, 2004*; *Cloninger, 2006*; *Cloninger, 2013*; *Cloninger, Svrakic & Svrakic, 1997*).

Accordingly, the dark character cube theorizes all eight possible combinations of high/low scores in Machiavellianism, narcissism, and psychopathy (see Fig. 1). One caveat here is that although the cube is based on Cloninger's theory of human character, we only make the assumption, based on the Dark Triad literature, that dark or malevolent character is ternary in nature. Nevertheless, far from the mixed patterns using the Big Five traits (e.g., *Jakobwitz & Egan, 2006*; *Paulhus & Williams, 2002*), the first attempt to use the Dark Cube as a model of malevolent character, suggested that Machiavellianism and psychopathy share a unified but unique non-agentic (low self-directedness) and non-communal (low cooperativeness) character; while narcissism has a unique character configuration expressed as high agency (high self-directedness). That being said, the Dark

[1] Equifinality: high scores in each one of the dark traits might have different antecedents, for example, individuals who are high in Machiavellianism might have different life events that explain their Machiavellian behavior. Multifinality: antecedent variables have different outcomes, for example, not all individuals who are extroverts end up scoring high in psychopathy and/or narcissism (*Garcia & Rosenberg, 2016*; see also *Cloninger & Zohar, 2011*).

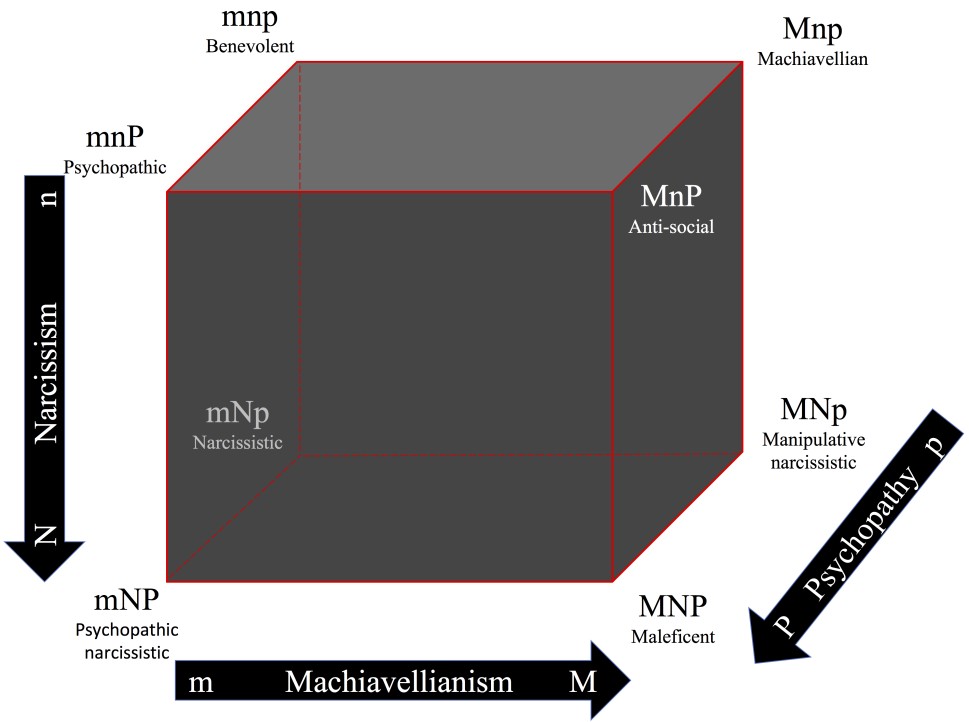

**Figure 1** **The Dark Cube as an analogy to Cloninger's character cube, showing all eight possible combinations of high/low scores in Machiavellianism, narcissism, and psychopathy.** Note: adapted with permission from CR Cloninger. The directions of the arrows represent higher values. M, high Machiavellianism; m, low Machiavellianism; N, high narcissism; n, low narcissism; P, high psychopathy; p, low psychopathy. Originally published in *Garcia & Rosenberg (2016)*.

[2] Recently, however, *Paulhus (2014)* has suggested everyday sadism as a fourth component, making the triad into a tetrad. It is plausible that future studies might find that enjoyment of cruelty against other human beings and animals is uniquely associated to the inability of transcend the self and feel part of the whole universe (*Garcia & Rosenberg, 2016*).

Triad seems to lack a dark trait that corresponds uniquely to a spiritual dimension of human character (i.e., self-transcendence)[2] (*Garcia & Rosenberg, 2016*). In the present study, we move forward the investigation of the Dark Cube as a model of malevolent character by investigating the associations between people's dark character and the Big Five traits. This study provides new data analysis for testing these associations by investigating differences between dark character profiles and openness, conscientiousness, extraversion, agreeableness, and neuroticism (i.e., OCEAN). Specifically, the use of the Dark Cube profiles (i.e., the combination of high/low in the three Dark Triad character traits) allows the comparison of individuals who differ in one dark character trait, while holding the other two constant.

## METHOD

### Ethical statement

After consulting with the Network for Empowerment and Well-Being's Review Board we arrived at the conclusion that the design of the present study (e.g., all participants' data were anonymous and will not be used for commercial or other non-scientific purposes) required only informed consent from the participants.

## Participants and procedure

Participants ($N = 330$) were recruited through Amazon's Mechanical Turk (MTurk; http://www.mturk.com/mturk/welcome) (for validation of MTurk as a data collection tool see among others *Rand, 2011*; *Buhrmester, Kwang & Gosling, 2011*). Participants were recruited on the criteria of being residents of the US and having English as their first language (parameters controlled through MTurk). All participants were informed that the survey was voluntary, anonymous, and that the participants could terminate the survey at any time. The MTurk workers received 50 cents (US dollars) as compensation for participating. Two control questions were added to the survey to control for automatic responses (e.g., "this is a control question, please answer "neither agree or disagree"). After taking away those who responded erroneously to one or both of the control questions ($n = 30$, 9.09% of all who participated), the final sample constituted 300 participants, 104 males (34.67%) and 196 females (65.33%), with an age $mean = 34.18$ years, $SD = 12.09$.

## Instruments

### The Short Dark Triad Inventory (*Jones & Paulhus, 2014*)

This instrument comprises 27 items, nine per each dark trait. Examples of the items are: "Most people can be manipulated" (Machiavellianism, *Cronbach's α = .78*), "People see me as a natural leader" (narcissism; *Cronbach's α = .75*), and "Payback needs to be quick and nasty" (psychopathy; *Cronbach's α = .74*). The items were rated on a 5-point *Likert* scale (1 = *strongly disagree*, 5 = *strongly agree*). The whole scale had a *Cronbach's α = .86*.

### The Big Five Inventory (*Benet-Martínez & John, 1998*)

This instrument comprises 44-item (5-point Likert scale: 1 = strongly disagree to 5 = strongly agree) that enables efficient assessment of the five personality dimensions: Openness (e.g., I see myself as a person who "is original, has new ideas"), Conscientiousness (e.g., I see myself as a person who "does things efficiently"), Extraversion (e.g., I see myself as a person "who is talkative"), Agreeableness (e.g., I see myself as a person who "has a forgiving nature"), and Neuroticism (e.g., I see myself as a person "who worries a lot"). Cronbach's α varied between .76 and .86 among traits.

## Statistical procedure

The sample was divided into subjects above (high) and below (low) the median[3] for each of the three dark traits: Machiavellianism (*median* = 3.00; M for high, m for low), narcissism (*median* = 2.67; N for high, n for low), and psychopathy (*median* = 1.78; P for high, p for low). Then the participants were grouped according to all the possible combinations of high and low dark trait scores to define the eight possible Dark Triad profiles: MNP "maleficent" ($n = 73$, 24.6%), MNp "manipulative narcissistic" ($n = 11$, 3.7%), MnP "anti-social" ($n = 29$, 9.8%), Mnp "Machiavellian" ($n = 30$, 10.1%), mNP "psychopathic narcissistic" ($n = 31$, 10.4%), mNp "narcissistic" ($n = 32$, 10.8%), mnP "psychopathic" ($n = 17$, 5.7%), and mnp "benevolent"[4] ($n = 74$, 24.9%). All analyses were conducted using SPSS version 24. We opted to conduct t-tests between the profiles because it permitted us to compare profiles that differed in one malevolent trait but were similar in the other two. Thus, allowing us to understand conditional relationships between each malevolent

[3] "The median is another way to measure the center of a numerical data set. A statistical median is much like the median of an interstate highway. On many highways, the median is the middle, and an equal number of lanes lay on either side of it. In a numerical data set, the *median* is the point at which there are an equal number of data points whose values lie above and below the median value. Thus, the median is truly the middle of the data set." *Rumsey (2017)*.

[4] In an earlier version of the Dark Cube model (*Garcia & Rosenberg, 2016*) the mnp profile was labeled "agreeable". After a well asserted commentary from one of the reviewers, in the first revision of the present paper, we opted to change the label to "benevolent" as recommended by the reviewer, which is in perfect contrast to the MNP or "maleficent" profile. We found this appropriate, since as pointed out by the reviewer the label "agreeable" posits agreeableness as an a priori anti-thesis of the Dark Triad, which can be confusing.

character and each OCEAN trait (cf. *Cloninger & Zohar, 2011*; *Schütz, Archer & Garcia, 2013*, who conducted the same procedure using the three character traits in Cloninger's model and well-being measures as the dependent variables). Moreover, since the dark traits are moderately correlated to each other, we found the $t$-test as a better solution that analysis of variance (cf. *Iacobucci et al., 2015a*; *Iacobucci et al., 2015b*, who showed that median splits, when accompanied by multicollinearity, can cause problems in the analysis of variance or in multiple regression).

## RESULTS

As a first analysis, we correlated participants' scores in the dark traits with their scores in the Big Five (see Table 1). The significant correlations between Machiavellianism and OCEAN were to conscientiousness ($r = -.16$, $p < .01$), agreeableness ($r = -.43$, $p < .01$), and neuroticism ($r = .19$, $p < .01$). Narcissism was associated to openness ($r = .17$, $p < .01$), extraversion ($r = .46$, $p < .001$), and neuroticism ($r = -.18$, $p < .01$). Finally, psychopathy was associated to conscientiousness ($r = -.31$, $p < .01$), extraversion ($r = .14$, $p < .05$), and agreeableness ($r = -.51$, $p < .001$). Importantly, the only associations above $\pm.20$[5] were Machiavellianism-agreeableness, narcissism-extraversion, and psychopathy-agreeableness. As a second analysis, we conducted the same analysis between the dark traits and OCEAN controlling for gender, since the dark traits differ between males and females (e.g., *Garcia, MacDonald & Rapp-Ricciardi, 2017*). Nevertheless, the correlations were almost similar (see Table 1).

As in earlier studies (e.g., *Cloninger & Zohar, 2011*; *Garcia & Rosenberg, 2016*), paired $t$-tests were performed to evaluate the conditional relationships between each of the Dark Triad and the Big Five traits. The comparisons investigated the effect of extremes of each Dark Triad trait when the other two were held constant (see Table 2 for the details). The only two clear associations were found between high narcissism and high extraversion and between high psychopathy and low agreeableness. These findings are in line with the correlation analyses above. However, the rest of the results were complex interactions and not necessarily in line with a unified view of the dark traits or simple linear associations between the Big Five and the Dark Triad traits.

## DISCUSSION

In the present study, we used the Dark Cube profiles to investigate conditional relationships between dark character and OCEAN. Specifically, the eight possible combinations of individuals' high/low scores in the three Dark Triad character traits were used to investigate differences in OCEAN between individuals who differ in one dark character trait, while holding the other two constant. In essence, our results showed that the relationship between one dark trait and OCEAN is valid only under certain conditions, that is, depending on individual scores in the other two dark traits. The only OCEAN traits associated to malevolent character in any condition or dark character combination were extraversion-narcissism, suggesting that an individual high in narcissism would independently of the other two dark traits always behave as an extrovert, and psychopathy-agreeableness,

[5] See *Ferguson (2009)* who suggested that an effect size of .20 as the recommended minimum effect size representing a practically significant effect for data in the social sciences. However, the debate of what constitutes a meaningful effect size is more complex than relegating it to a .20 level (e.g., *McGraw, 1991*; *Rosenthal, 1991*; *Rosenthal & Rubin, 1982*; *Strahan, 1991*; *Thompson & Schumacker, 1997*).

**Table 1** Correlations, *means*, *standard deviations*, and *Cronbach's α* for Dark Triad and Big Five traits.

| | | Big Five | | | | | Dark Triad | | |
|---|---|---|---|---|---|---|---|---|---|
| | | **O** | **C** | **E** | **A** | **N** | **M** | **Narc** | **P** |
| Big Five | Openness (O) | – | .14* | .23** | .12* | −.08 | .01 | .20** | −.01 |
| | Conscientiousness (C) | .14* | – | .16** | .43*** | —**** | −.15* | .06 | −.31*** |
| | Extraversion (E) | .21*** | .16** | – | .22*** | −.32*** | −.07 | .46*** | .13* |
| | Agreeableness (A) | .13* | .44*** | .21*** | – | −.50*** | −.41*** | −.07 | −.50*** |
| | Neuroticism (N) | −.06 | −.45** | −.31** | −.43** | – | .27*** | −.12* | .28*** |
| Dark Triad | Machiavellianism (M) | .01 | −.16** | −.06 | −.43** | .19** | – | .32*** | .48*** |
| | Narcissism (Narc) | .17** | 0.04 | .46*** | −0.11 | −.18** | .35*** | – | .38*** |
| | Psychopathy (P) | −.03 | −.31** | .14 * | −.51*** | 0.11 | .50*** | .44*** | – |
| | *Means* and *sd* (±) | 36.72 ± 6.46 | 34.21 ± 6.05 | 23.97 ± 6.97 | 34.55 ± 5.82 | 22.93 ± 7.13 | 2.98 ± 0.71 | 2.72 ± 0.69 | 1.82 ± 0.60 |
| | *Cronbach's α* | .76 | .79 | .86 | .78 | .86 | .78 | .75 | .74 |

**Notes.**

White cells mark bivariate correlations between dark traits and ocean; grey cells mark partial correlations between dark traits and ocean controlling for gender; black cells mark significant correlations between dark traits and OCEAN (both bivariate and partial).

Garcia and González Moraga (2017), *PeerJ*, DOI 10.7717/peerj.3845

**Table 2  Results from the *t*-tests analyses for each Dark Triad character trait for openness, conscientiousness, extraversion, agreeableness, and Neuroticism (OCEAN).** The black cells indicate significant results.

| | | Openness | | | Conscientiousness | | | Extraversion | | | Agreeableness | | | Neuroticism | | |
|---|---|---|---|---|---|---|---|---|---|---|---|---|---|---|---|---|
| | | *t* | *p* | *d* | *t* | *p* | *d* | *t* | *p* | *d* | *t* | *p* | *d* | *t* | *p* | *d* |
| Machiavellianism | MNP vs. mNP | −0.27 | .79 | −0.07 | 1.16 | .25 | 0.25 | −0.23 | .82 | −0.05 | −0.86 | .39 | −0.17 | −1.33 | .19 | −0.29 |
| | MNp vs. mNp | 0.08 | .94 | 0.03 | −0.49 | .62 | −0.16 | 0.50 | .62 | 0.18 | −1.13 | .27 | −0.30 | 0.02 | .98 | 0.01 |
| | MnP vs. mnP | −1.01 | .32 | −0.34 | −0.06 | .95 | −0.02 | −1.63 | .11 | −0.48 | −2.63 | .01; | −0.80 | 1.08 | .28 | 0.34 |
| | Mnp vs. mnp | 0.96 | .34 | 0.22 | −2.27 | .03; | −0.41 | −2.01 | .05; | −0.49 | −3.95 | <.001 | −0.81 | 2.01 | .05 | 0.45 |
| Narcissism | MNP vs. MnP | 2.28 | .02 | 0.49 | 2.95 | <.001 | 0.68 | 5.08 | <.001 | 1.18 | 1.80 | .07 | 0.38 | −2.32 | .02 | −0.51 |
| | MNp vs. Mnp | 0.70 | .49 | 0.24 | 1.29 | .20 | 0.52 | 3.47 | <.001 | 1.33 | 1.78 | .08 | 0.60 | −2.46 | .02 | −0.78 |
| | mNP vs. mnP | 0.57 | .57 | 0.16 | 1.21 | .23 | 0.41 | 2.06 | .04 | 0.63 | −0.47 | .64 | −0.15 | 0.37 | .71 | 0.11 |
| | mNp vs. mnp | 2.03 | .04 | 0.52 | 0.98 | .33 | 0.21 | 2.31 | .02 | 0.59 | 0.84 | .40 | 0.21 | −2.11 | .04 | −0.50 |
| Psychopathy | MNP vs. MNp | −0.74 | .46 | −0.25 | −1.53 | .13 | −0.49 | 0.08 | .94 | 0.03 | −2.10 | .04 | −0.67; | 1.30 | .20 | 0.43 |
| | MnP vs. Mnp | −1.59 | .12 | −0.47 | −2.37 | .02 | −0.60 | 0.61 | .54 | 0.15 | −2.47 | .02 | −0.64 | 0.24 | .81 | 0.06 |
| | mNP vs. mNp | −0.44 | .66 | −0.09 | −3.85 | <.001 | −0.88 | 0.98 | .33 | 0.25 | −4.48 | <.001 | −1.02; | 2.98 | <.001 | 0.70 |
| | mnP vs. mnp | 0.41 | .68 | 0.12 | −4.43 | <.001 | −0.86 | 0.59 | .55 | 0.19 | −2.56 | .01 | −0.64 | 0.58 | .57 | 0.15 |

**Notes.**

*d*, *Cohen's d.*; M, high Machiavellianism; m, low Machiavellianism; N, high narcissism; n, low narcissism; P, high psychopathy; p, low psychopathy; MNP, "maleficent"; MNp, "manipulative narcissistic"; MnP, "anti-social"; Mnp, "Machiavellian"; mNP, "psychopathic narcissistic"; mNp, "narcissistic"; mnP, "psychopathic"; mnp, "benevolent".

suggesting that an individual high in psychopathy would independently of the other two dark traits always behave disagreeable. In other words, our findings might (1) explain the mixed and inconsistent associations between dark traits and OCEAN and (2) suggest that at least for individuals high in narcissism or high in psychopathy, introvert and agreeable behavior, respectively, might depend of external conditions rather than her/his own character combination (cf. *Birkás et al., 2016*, who show that anxiety is related to malevolent character under specific conditions). Next, we detail and describe the significant associations between the Dark Triad and each OCEAN trait.

High levels of Machiavellianism were associated to low levels of conscientiousness, low levels of extraversion, low levels of agreeableness, and high levels neuroticism only when both narcissism and psychopathy were low (Mnp vs. mnp). In other words, these results suggest that, as long as the other two malevolent traits are low, high levels of Machiavellianism would lead to low sense of competence, disorderliness, low dutifulness, low self-discipline (i.e., low levels of conscientiousness), low degree of displayed affection, low experience of positive emotions, low need of social affiliation (i.e., low levels of extravertion), low levels of trust in others, low degrees of sincerity, unhelpfulness, aggressive behavior, arrogance, low empathy (i.e., low levels of agreeableness), and proneness to worry, rumination, hostility, sadness, hopelessness, impulsiveness, and sensitivity in social situations (i.e., high levels of neuroticism). In addition, low levels of agreeableness were also associated to high Machiavellianism when psychopathy was also high but narcissism was low (MnP vs. mnP). In short, these results are partially in line with past research, however, only under the following conditions: both narcissism and psychopathy are low and in the case of agreeableness also when narcissism was low at the same time that psychopathy was high. Hence, for other combinations, high levels of Machiavellianism do not show a straightforward relationship to OCEAN. This is surprising, because most research suggest that low agreeableness is the common core of the dark traits (e.g., *Jakobwitz & Egan, 2006*; *Paulhus & Williams, 2002*). Even the correlation analyses in our study suggested a high Machiavellianism-low agreeableness correlation. Our analyses, however, suggest that under certain conditions individuals high in Machiavellianism might or might not develop agreeableness.

With the exception of its relation to extraversion, the results with regard to narcissism presented a more complex interaction with OCEAN. For example, narcissism was associated to high openness when both Machiavellianism and psychopathy were also high (MNP vs. MnP) and when both Machiavellianism and psychopathy were low (mNp vs. mnp). High levels of narcissism were associated to low levels of neuroticism in most of the cases but not associated at all when Machiavellianism was low and psychopathy was high (mNP vs. mnP). That is, individuals with a "psychopathic narcissistic" profile (mNP) might or might not have a proneness to worry, rumination, hostility, sadness, hopelessness, impulsiveness, and sensitivity in social situations (i.e., high levels of neuroticism; see for example the results mNP vs. mNp in Table 2). Interestingly, psychopathy has been found to correlate negatively to neuroticism (e.g., *Paulhus & Williams, 2002*) and negative affect (*Love & Holder, 2014*), which is almost synonymous with neuroticism (e.g., *Tellegen, 1993*; *Watson, Clark & Tellegen, 1988*). Nevertheless, some studies have not replicated the link

high psychopathy-low neuroticism (*Veselka, Schermer & Vernon, 2012*; *Garcia et al., 2015*). The findings presented here, however, suggest that both high and low neuroticism might be found in individuals high in psychopathy, hence suggesting the probability of both an emotionally stable and a emotionally instable psychopath. For instance, recent research suggests that anxiety, a state usually experienced by individuals high in neuroticism, is related to malevolent character under specific conditions (*Birkás et al., 2016*). In addition, high levels of narcissism were associated to high levels of conscientiousness when both Machiavellianism and psychopathy were high at the same time (MNP vs MnP). At this point, as long as the other two traits are high, high levels of narcissism are associated to high openness (i.e., proneness to imagination, appreciation of beauty, receptiveness to emotions, novelty seeking, and inquisitiveness), high conscientiousness, high extraversion, and low neuroticism. In other words, under these conditions, individuals with a ''maleficent'' profile (MNP) are better fitted to manipulate interpersonal relations with what might be interpreted as more social and adaptive abilities. We see this as manipulation and not true cooperative character, since the ''maleficent'' profile has been associated to low levels of two important measures of individuals goals and values: self-directedness (i.e., self-acceptance, self-fulfillment, goal-directedness) and cooperativeness (i.e., helpfulness, empathy, tolerance towards others, kindness) (*Garcia & Rosenberg, 2016*).

Finally, as earlier stated, psychopathy was the only dark trait with a clear association to agreeableness that was consistent with the general idea of the Dark Triad having a common core: unagreeablness (e.g., *Garcia et al., 2015*; *Garcia & Rosenberg, 2016*; *Kajonius et al., 2016*). As in past studies using, high levels of psychopathy were associated to low levels of conscientiousness in most of the cases. The only exception was when both Machiavellianism and narcissism were high (MNP vs. MNp). That is, an individual high in psychopathy might or might not be conscientious when the other two malevolent character traits are high (see for example the results between MNP vs. MnP in Table 2). In addition, as detailed earlier, high levels of psychopathy were associated to high levels of neuroticism when Machiavellianism was low and narcissism was high (mNP vs. nNP).

## Limitations and recommendations for future venues

The most obvious limitation is that our study was cross-sectional, thus, no causal effects can be discerned or established. Another limitation was that females represented 65.33% of the sample and it is possible that results have been affected by gender differences. However, the correlation analyses controlling for gender did not show any discrepancies that were noteworthy. Some aspects related to the use of MTurk, such as, workers' attention levels, cross-talk between participants, and the fact that participants get remuneration for their answers, might also have affected the results (*Buhrmester, Kwang & Gosling, 2011*). Nevertheless, a large quantity of studies show that data on psychological measures collected through MTurk meets academic standards, is demographically diverse, that payment does not seem to affect data quality, and also that health measures show satisfactory internal as well as test–retest reliability (*Buhrmester, Kwang & Gosling, 2011*; *Horton, Rand & Zeckhauser, 2011*; *Shapiro, Chandler & Mueller, 2013*; *Paolacci, Chandler & Ipeirotis, 2010*).

It is also plausible to argue that dichotomizing into groups that are classified as being low or high on traits will likely cause loss of power that is equivalent to the loss in sample size (e.g., *MacCallum et al., 2002*). For instance, some of the profiles in the present study contained a relatively low number of individuals, which might lead subsequent analyses to be less likely to find support for the hypotheses (i.e., Type II errors; *Humphreys, 1978*; *Lagakos, 1988*). Thus, the present results need to be replicated using large enough sample sizes. However, the reader should have in mind that, despite median splits making our analyses more conservative, we found significant differences in our sample. That being said, since median splits distort the meaning of high and low, it is plausible to criticize the validity of this approach to create the profiles—scores just-above and just-below the median become high and low by arbitrariness, not by reality (*Schütz, Archer & Garcia, 2013*; *Garcia, MacDonald & Archer, 2015*). That is, there still is a risk that dichotomizing the dark traits might have led to spurious main effects (cf. *MacCallum et al., 2002*; *Maxwell & Delaney, 1993*; *Fitzsimons, 2008*). Nevertheless, there is recent evidence of the statistical robustness and valid use of median splits (*Iacobucci et al., 2015a*; *Iacobucci et al., 2015b*) and also evidence of median splits being as reliable as cluster methods (*Garcia, MacDonald & Archer, 2015*). In short, although there is a risk for misleading results when using median splits, stating that median splits produce inferior analytic conclusions is a simplification and misconception of the real issue (*Iacobucci et al., 2015a*; *Iacobucci et al., 2015b*).

In addition, others might argue that the shared variance of the dark traits is as important as their unshared variance. Nonetheless, some of the problems with current Dark Triad research are the unreliable correlations with available models (*Veselka, Schermer & Vernon, 2011*) and the difficulty on differentiating them from each other (*Garcia & MacDonald, 2017*). The later probably reflects operationalization problems (*Garcia & Rosenberg, 2016*; *Garcia, MacDonald & Rapp-Ricciardi, 2017*; *Garcia et al., 2017e* ; *Kajonius et al., 2016*; *Persson, Kajonius & Garcia, 2017a*; *Persson, Kajonius & Garcia, 2017b*) that are beyond the scope of the present paper. The former, however, is directly addressed in our study, since the model presented here allows us to conduct analysis of personality as a complex adaptive system–a system that allows the individual to adapt to internal (i.e., different character combinations, which is the focus of our study) and external conditions (e.g., life events and situations) (cf. *Cloninger, 2004*). In other words, the present study focus on personality as being non-linear, which is, as far as we know, a new approach for investigating the dark traits but common in the study of human character (*Cloninger, 2004*). Indeed, others have argued that from a person-centered framework personality dimensions within the individual can be seen as interwoven components with whole-system properties (*Cloninger, 2004*; *Bergman & Wångby, 2014*). In fact, the Dark Cube allowed us to investigate both shared and unshared variance. For example, we could affirm that low agreeableness is associated to psychopathy independently from the other two malevolent character traits, associated to Machiavellianism only under certain conditions (i.e., for individuals with either an "anti-social" or MnP profile and a "Machiavellian" or Mnp profile), but not associated to narcissism under any condition. In other words, suggesting shared and unshared variance in agreeableness among individuals with different dark character profiles.

At the end, however, we only present a model of malevolent character based on Cloninger's biopsychosocial model of personality. Our scientific model represents phenomena (in this case the Dark Triad or malevolent character) in a logical but simplified way (cf. *Apostel, 1960*; *Atkinson, 1960*; *Chakravartty, 2010*; *Toon, 2010*). We (*Garcia & Rosenberg, 2016*) found reasonable to suggest the Dark Cube as a model of malevolent character based on the non-linear nature of personality (*Cloninger, 2004*) and the assumption of an actual Dark Triad in most of the literature studying dark malevolent traits. Only using the Dark Cube model and other person-centered methods, such as cluster analysis (*Garcia & MacDonald, 2017*; *Kam & Zhou, 2016*) can we come to an agreement on its usefulness in the development of a Dark Triad Theory.

## Concluding remarks

In contrast to previous notions of disagreeableness being the core of the Dark Triad, our study suggest that this might (1) be true for Machiavellianism only when the other two dark character traits are low or when narcissism is low at the same time that psychopathy is high, (2) not be true for narcissism, and (3) only be totally true for psychopathy. In this vein, researchers have suggested a "Dark Dyad" either by excluding narcissism (i.e., Dark Dyad = Machiavellianism and psychopathy; *Egan, Chan & Shorter, 2014*) or by suggesting an amalgamated anti-social trait (i.e., Machiavellianism + psychopathy) and narcissism (*Garcia & Rosenberg, 2016*; *Kajonius et al., 2016*). Some of the arguments to this stance, besides factor analyses studies, are results showing that the General Factor of Personality[6] is negativity associated to Machiavellianism and psychopathy but it is not significantly associated to narcissism (*Kowalski, Vernon & Schermer, 2016*). At first sight, this insight favors a "Dark Dyad" including only Machiavellianism and psychopathy. However, the lack of linear associations between narcissism and the General Factor of Personality does not rule out the results found here. Our results, for example, suggest that in certain conditions only narcissism was associated to high levels of openness to experience, high conscientiousness (antagonist with Machiavellianism and psychopathy), high extraversion (antagonist with Machiavellianism) and low neuroticism (antagonist with Machiavellianism and psychopathy) (see also *Kowalski, Vernon & Schermer, 2016*; *Egan, Chan & Shorter, 2014*; *Veselka, Schermer & Vernon, 2011*; *Vernon et al., 2008*). In addition, the lack of association between narcissism and agreeableness seen here should not be interpreted to suggest that individuals high in narcissisms are cooperative, helpful, and empathic. After all, individuals high in narcissism tend to manipulate others to gain self-validation with no regard to who they might hurt in doing so (*Watson et al., 1984*). Our suggestion is that it is too early to rule out a "Dark Dyad" including an amalgamated anti-social trait (i.e., Machiavellianism + psychopathy) and narcissism. After all, this pattern has been discerned in two studies using Item Response Theory analyses and two different measures of the dark character traits (*Kajonius et al., 2016*; *Persson, Kajonius & Garcia, 2017a*; *Persson, Kajonius & Garcia, 2017b*). That being said, a Dark Triad Theory would benefit of a person-centered approach around a biopsychosocial model (cf. Cloninger's biopsysocial model of personality, *Cloninger, 2004*). The Dark Cube with its eight dark

[6]The General Factor of Personality is a reduction of the Big Five traits into one single dimension (*Kowalski, Vernon & Schermer, 2016*). A person who scores high in this personality dimension is characterized as having a blend of socially desirable personality traits: high extraversion, low neuroticism, high openness to experience, high conscientiousness, and high agreeableness (*Musek, 2007*).

profiles is suggested here as tool to shade light on the mixed and inconsistent linear associations in the Dark Triad literature.

*"Nonlinear interactions almost always make the behavior of the aggregate more complicated than would be predicted by summing or averaging."*

*John Henry Holland (Holland, 1995)*

### Funding

The development of this article was funded by a grant from the Swedish Research Council (Dnr. 2015-01229). The funders had no role in study design, data collection and analysis, decision to publish, or preparation of the manuscript.

### Grant Disclosures

The following grant information was disclosed by the authors:
Swedish Research Council: 2015-01229.

### Competing Interests

The authors declare there are no competing interests. Danilo Garcia is the Head of Research of the Blekinge Center of Competence, which is the Blekinge County Council's research and development unit. The Center works on innovations in public health and practice through interdisciplinary scientific research, community projects, and the dissemination of knowledge in order to increase the quality of life of the habitants of the county of Blekinge, Sweden. Danilo Garcia is, together with Professor Trevor Archer and Associate Professor Max Rapp Ricciardi, the lead researcher of the Network for Empowerment and Well-Being.

### Author Contributions

- Danilo Garcia conceived and designed the experiments, performed the experiments, analyzed the data, contributed reagents/materials/analysis tools, wrote the paper, prepared figures and/or tables, reviewed drafts of the paper.
- Fernando R. González Moraga wrote the paper, reviewed drafts of the paper.

### Human Ethics

The following information was supplied relating to ethical approvals (i.e., approving body and any reference numbers):

After consulting with the Network for Empowerment and Well-Being's Review Board we arrived at the conclusion that the design of the present study (e.g., all participants' data were anonymous and will not be used for commercial or other non-scientific purposes) required only informed consent from the participants.

### Data Availability

The dataset is available as Supplementary Information.

## Supplemental Information

Supplemental information for this article can be found online at http://dx.doi.org/10.7717/peerj.3845#supplemental-information.

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
