# Peer review of "The Dark Cube: dark character profiles and OCEAN"

_PeerJ, doi:10.7717/peerj.3845_

## Round 0.1 · original submission · Major Revisions

· Academic Editor

Major Revisions

Your manuscript received reviews requesting major revisions. The process of re-review will be facilitated if you respond to the reviewers on an item-by-item basis and specify where changes have been made (yellow highlighting of revised text would be helpful). Be sure that reviewers' requests are addressed in the text, and not merely in the response to reviewers.

·

Basic reporting

The paper is a standard descriptive study with all appropriate details included.

Experimental design

While not experimental, the research design is clear. I am troubled with the assumptions made in the cube design. While the authors notes the limitations of median splits in brief I think more needs to be said about problems with assuming traits are categorical as opposed to continuous and how this approach is better than a latent variable approach. The authors repeatedly allude to the unification or dark dou but it is unclear whether the present analysis is best here. The important point that might be missed in relation to the former is that it is the SHARED variance not the unshared variance that is of interest to many DT researchers. In reference to the second point, the two factor solutions have reliably been detected in very large samples which introduce more correlated errors--errors that M and P people may make systematically more than N people.

Validity of the findings

The findings are purely descriptive in nature. This study does not advance any DT theory is not one. The use of the B5 is a descriptive tool that allows us to understand the DT traits. As the B5 have become the language of all personality psychology, framing the DT in terms of the B5 helps others understand what is meant by the traits. However, all we really have here is correlated semantic agreement. The study does not advance any theoretical model and instead adds to the noise of B5 studies on the DT. While the authors have adopted a novel method, the results should not be oversold as resolving any conflict or question.

Additional comments

I find it notable that you write an entire paper on the DT and not cite me one time. This is especially problematic as I was the one who first postulated the presence of a DT factor; a finding that has been replicated both by my work but also other work by Figueredo et al. (2015; Evolutionary Psychology). The primary limitation of this paper is that is lacks any real thoeretical heft. It is completely descriptive in nature but the authors seem to overstep their mark here.

·

Basic reporting

The manuscript ’The Dark Cube: Dark Character Profiles and OCEAN’ by D. Garcia and F.R.G. Moraga addresses an important issue in the research of the Dark Triad (DT). Namely they argue to investigate the possible combinations of high and low DT traits in relation to the Big Five. This line of research is of high importance because the unity of DT traits has been questioned by several authors. The current study tries to make good for the lack of consistent relationships between DT and the Big Five throughout the literature.
The introduction gives a concise but clear outline of the problem and of previous research in the field. However, I found some of the phrasing somewhat misleading throughout the manuscript. I will list all three of them here.
(1) Labelling the mnp profile (low Mach, low narcissism, low psychopathy) as ‘agreeable’ might be confusing, because it posits agreeableness as an a priori anti-thesis of the DT (which is in fact the case in many studies). I would suggest to label mnp as ‘beneficent’ in contrast to the label of MNP (i.e. maleficent).
(2) Authors highlight the importance of investigating non-linear relationships between DT and OCEAN. I found it also misleading, because linear relationships are investigated (comparing two groups can only be linear). At the same time, I agree with authors about the importance of investigating conditional or moderated relationships. I would suggest two use on of the previously mentioned terms instead of non-linear.
(3) Although it has been already published (Garcia and Rosenberg, 2016) – and I can accept if authors chose to stick with their original phrasing –, labelling the character cube by Cloninger as ‘light’ might be also confusing. Cloninger’s TCI measure is frequently used in the assessment of personality disorders that are mainly based on character traits (especially SD and C). Simply omitting ‘light’ would easily solve this problem.
Moreover, please spell-check the manuscript throughout. E.g., p. 11 row 3 would go correctly ‘… is negatively associated to …’.

Experimental design

The study used a self-report cross-sectional design. The limitations resulting from this design should be mentioned somewhere in the Limitations and concluding remarks section. Authors should also report, which statistical software they used for the analyses.

Validity of the findings

The findings in their current form cannot be considered as valid. I will list my concerns below in detail.
(1) Dividing the sample into groups representing different Dark Cube profiles leaves the authors with two problematic groups. MNp consists of 11 participants and mnP consists of 17 participants. Using these groups in any comparison makes the validity of the results highly questionable. Suggested solution: authors should at least add a dimensional test of moderated moderation (or three-way interaction) for DT traits. This would be the same statistical approach as in Garcia and Rosenberg (2016). Using modules like e.g., PROCESS for SPSS would also enable authors to trace the conditional effects of DT traits on OCEAN.
(2) Since gender distribution is uneven in the sample and both DT and OCEAN are gender-sensitive, authors should at least report the gender distribution for each profile group. They might be also willing to control for effects of gender (and perhaps age) in statistical analyses.
(3) Although it might not fit in one table, but somehow group descriptives (M and SD) should be reported, maybe along with Cohen’s ds as indices of power.

Additional comments

I would be really happy to see your manuscript published, because it could stir up the water in DT research. However, half of the sample categorized either as mnp or as MNP leaves me unpersuaded whether it is correct to talk about a Dark Cube. This distribution shows that M, N and P are highly correlated, whereas a cube has three orthogonal dimensions. Maybe establishing national standards for SD3 on a representative sample would help with categorizing anyone over 50 T as high on a trait and equal to or below 50 as low on a trait. It would give us a realistic picture about the proportion of any profile in the population (I would be really sad if – as your sample shows – we would have 25 % of the population as maleficent).

Reviewer 3 ·

Basic reporting

For clarity purposes the grammar and spelling should be checked. It would have been also really helpful, if the data and the code were attached. Without those it makes it hard to comprehend the outcomes of the paper.
The introduction makes really clear that personality can be understood as a complex interplay of traits. But the authors fail to make really clear what the advantages of the dark cube actually are, and how introducing personality types can account for that complexity.
While checking some citations I came across some problems also. A paper by Lee & Ashton (2013) is used to give an example where a certain correlation pattern between the Dark Triad and the Big Five can actually be found. This article doesn’t even contain measures of the Dark Triad, but it is an article about validity of the HEXACO and NEO-FFI, so it is not relevant to the topic. And just for example, another paper used to underscore the inconsistency of correlation patterns between the Dark Triad and the Big Five by Jakobwitz & Egan (2006), used different measures of those personality traits than the paper it was compared to (Paulhus & Williams, 2002). What I mean with that: differences in the correlation patterns can stem from a lot of different sources. Not mentioning or discussing those different sources seems unscientific.
The introduction doesn’t go enough into detail about the concept and the advantages of the Dark Cube, and needs to reassess the literature itself, as well as the way those publications are used.
The structure of the article does not conform to an acceptable format. Results and discussion should be handled separately. It is very important to first just show the results without any interpretation, and then interpret those.

Experimental design

A big part of my concerns with this paper is about the methods, and statistical analyses used. There is no explanation given in the method or result section why certain analyses where used. I would have liked to see for example an explanation given to why the median but not the mean was used to split the groups (as well as for other analyses: why a t-test?). This would support clarity and transparency. The reliability measures given for the Short Dark Triad (SD3) are not sufficient, please provide an internal consistency score for the whole measure also. It is not made clear, if the participants were speaking Spanish or not. Given the Spanish BFI measure I would think so, but there is no Spanish SD3 measure cited or mentioned. Please provide information about this issue.
I understood this paper as a way of showing researchers who work with the dark traits a new and different way of understanding and analysing the Dark Triad. For that it would have been important that the authors also show the actual correlations they found between the Dark Triad and Big Five, and compare them to the way they analysed. Not showing those results is highly problematic.
Some other information about statistical analyses that I missed were how the authors accounted for different sample sizes for the t test. Also there were no effect sizes provided, which are just as important as significance levels.
The explanation given in the discussion why it is statistically sound to divide 8 different personality types by the median of a normally distributed trait is not sufficient at all.
The discussion part doesn’t really focus on trying to find explanations why the results are the way they are. One really interesting result (that Agreeableness seems to be less connected to the dark traits than expected) was just mentioned and discussed at all, even though it is a highly controversial point to be made.

Validity of the findings

Validity of findings is already discussed in first two parts of review.

---

## Round 0.2 · accepted · Accept

· Academic Editor

Accept

I am pleased to inform you that your manuscript has been judged scientifically suitable for publication. Thank you for your contribution.

·

Basic reporting

The revised manuscript uses professional English. Introduction and discussion are provided with sufficient references and context. Article structure is according to standards, figure and tables are ok. The submission is self-contained.

Experimental design

Original research fitting into the Aims and Scope of PeerJ. Research question is well defined, addresses an important gap in Dark Triad Research. Investigation is in line with APA ethical standards. Method is described in sufficient detail for replication.

Validity of the findings

Data are robust and statistically sound. Conclusions are well stated. Speculations are clearly identified as such.

Additional comments

I have all my comments answered in sufficient form.